# Influence of a Novel Surface of Bioactive Implants on Osseointegration: A Comparative and Histomorfometric Correlation and Implant Stability Study in Minipigs

**DOI:** 10.3390/ijms20092307

**Published:** 2019-05-09

**Authors:** Manuel M. Romero-Ruiz, Francisco Javier Gil-Mur, José Vicente Ríos-Santos, Pedro Lázaro-Calvo, Blanca Ríos-Carrasco, Mariano Herrero-Climent

**Affiliations:** 1Advanced Periodontics, Facultad de Odontología, Universidad de Sevilla, C/Avicena S/N, 41009 Seville, Spain; mmromero@infomed.es (M.M.R.-R.); pedro@lazarocalvo.com (P.L.-C.); brios@us.es (B.R.-C.); 2Technological Health Research Center, Biomaterials of the Faculties of Medicine and Dentistry, International University of Cataluña, 08034 Barcelona, Spain; xavier.gil@uic.es; 3Porto Dental Institute, 4150-518 Oporto, Portugal; dr.herrero@herrerocliment.com

**Keywords:** implant surface, Bioactive Implants, Oral implants, Histomorfometric analytic, minipigs

## Abstract

Purpose: The objective of this study was to assess the influence of a novel surface of dental implants (ContacTi^®^) on the osseointegration process in a minipig model. The surface was compared with other existing surfaces on the market (SLA^®^ and SLActive^®^) by employing bone implant contact analysis (BIC) and implant stability. Method: Twelve minipigs were used with prior authorisation from an ethics committee. Three types of surfaces were tested: SLA^®^ (sand-blasted acid-etched titanium), SLActive^®^ (same but hydrophilic, performed under a nitrogen atmosphere), and ContacTi^®^ (alumina particle bombardment of titanium, bioactivated when treated thermochemically) in 4.1 mm × 8 mm implants with internal connection and a polished neck. Twelve implants of each surface type (*N* = 36) were placed, sacrificing 1/3 of the animals at 2 weeks of placement, 1/3 at 4 weeks and the remaining 1/3 at 8 weeks. Numerical variables were compared with Analysis of Variance, and the correlation between ISQ and BIC was established with the Spearman’s rank correlation coefficient. Results: SLActive^®^ and ContacTi^®^ surfaces showed elevated osteoconductivity at 4 weeks, maintaining a similar evolution at 8 weeks (large amount of mature lamellar tissue with high maturity and bone quality). The SLA^®^ surface showed slower maturation. The ISQ values in surgery were elevated (above 65), higher at necropsy and higher at 4 and 8 weeks in the SLA^®^ group than in the other two (SLActive^®^ and ContacTi^®^). No significant correlation was found between ISQ and BIC for each implant surface and necropsy time. Conclusion: The three surfaces analysed showed high RFA and BIC values, which were more favourable for the SLActive^®^ and ContacTi^®^ surfaces. No statistical correlation was found between the RFA and BIC values in any of the three surfaces analysed.

## 1. Introduction

Modern implantology has advanced in the development of new implant surfaces that can achieve a better and faster osseointegration. In recent years, as a result of the application of different physical and biochemical treatments, a new generation of so-called bioactive surfaces has emerged; these surfaces are capable of accelerating cellular differentiation in the osteogenic pathway, improving upon the results of their predecessors [1]. The improvement in the properties of the implant surfaces is even more relevant in cases of poor bone quality or in treatments in which shortened loading times are the intended outcome [2].

Among these surfaces is SLActive^®^ (Institut Straumann AG, Basel, Switzerland), which has shown excellent results in both experimental [3] and clinical studies [4,5]. Recently, a new surface, ContacTi^®^ (Klockner Implant System, SOADCO, Andorra), obtained through a two-step titanium treatment, has been introduced: The surface is blasted with alumina particles to achieve an optimal micro-roughness for the adhesion, proliferation, and differentiation of human osteoblast cells and a subsequent treatment (alkaline immersion and heat and biomimetic treatments) that confer bioactivity [6]. This technique was tested in vitro: a layer of hydroxyapatite formed on the surface when the implant was exposed to a simulated body fluid, which improved the osteoconductivity of the implant, accelerated early bone healing by promoting rapid protein adsorption and accelerated the union between the implant and the bone [7,8,9].

The adequate stability of the implant in the different phases of the osseointegration process is essential for long-term success in this type of treatment. Several methods have been utilised to measure the stability of the implant, with the resonance frequency analysis (RFA) using the Osstell device (Osstell AB, Goteborg, Sweden) commonly used today [10,11,12]. This method is based on the measurement of the oscillation frequency of the implant in the bone, induced by a magnetic pulsing stimulus such that the said frequency is amplified, analysed and transformed by the device into units called ISQ values (implant stability quotient) in a range between 1 and 100 [13,14]. In vitro studies have shown that these ISQ values increase with the stiffness of the bone-implant interface, with an inverse correlation observed between these values and the lateral displacement of the implant subjected to loads [11,13]. RFA has proven to be a useful tool for assessing the evolution of implant osseointegration because it allows for a clinical measurement of the quality of the bone-implant interface immediately after insertion (primary stability), as well as biological anchorage or secondary stability that occurs in the following weeks [15].

Because changes in implant stability values reflect the stiffness of the implant-bone union, it is logical that these values should correlate with data obtained in histomorphometric studies, especially with the bone-implant contact (BIC value) percentage. In this sense, various correlation studies have been performed between both variables (ISQ and BIC values) based on the hypothesis that an increase in bone density, and therefore, the percentage of bone-implant contact would lead to an increase in the stiffness of the union, which would translate into an increase in implant stability [16]. However, published studies have shown contradictory results. Thus, some authors, such as Ito et al. [17] in a study on minipig tibias or Rocci et al. [18] in implants removed in humans, have found no relationship between RFA and BIC values. However, other authors, such as Gedrange et al. [19] or Nkenke et al. [20], have found a statistical correlation between both values in human cadaver studies.

The purpose of this study is to evaluate the influence of a new implant surface, ContacTi^®^ (Klockner Implant System, SOADCO, Andorra), on the osseointegration process in a minipig model and to compare this surface with other existing surfaces available on the market by utilising histomorphometric (BIC) and implant stability (ISQ) data. This study will also evaluate whether a correlation exists between both values.

## 2. Results

The surgical procedures and the healing phase occurred without incident for all implants. No infectious processes, wound dehiscence or implant loss were recorded at the time of sacrificing the animal.

In total, 36 implants were placed in 12 pigs, so each animal received 3 implants, one for each surface under study. The animals were distributed in three groups of 4 pigs each; group 1 was euthanized at 2 weeks after surgery; group 2 was euthanized at 4 weeks; and group 3 was euthanized at 8 weeks after surgery. All of the samples studied exhibited a normal macroscopic anatomy, showing normal bone contact with all implants, without signs of fibrosis or inflammation.

### 2.1. Histological Analysis

For the three time points in the study, the histological samples showed the interaction between the surfaces and the surrounding bone. Thus, at two weeks, the milling lines and the existence of primary contact with the turns were observed, such that the ContacTi^®^ and SLActive^®^ surfaces already showed signs of new immature bone, with osteogenesis occurring at the points of contact, presenting as new bone formation between the implant and the surrounding mature bone. At 4 weeks, the ContacTi^®^ and SLActive^®^ surfaces showed elevated osteoconductivity, highlighting the presence of significant amounts of immature bone tissue growing from the surface of the implant. A significant amount of bone remodelling was also observed (Figure 1).

Both surfaces underwent a similar evolution at 8 weeks, showing a large amount of lamellar tissue as well as strong osteoconductivity. There was a significant amount of mature lamellar tissue and minimal vascular space; thus, there was a high level of maturity and bone quality, indicating an advanced state of bone remodelling. The SLA^®^ surface showed a slower rate of maturation (Figure 1 and Figure 2).

### 2.2. Resonance Frequency Analysis (RFA)

RFA measurements were performed immediately after placement of the implants and at the time of sacrifice, i.e., at two, four or eight weeks according to the group. All mean ISQ values on the day of surgery were high, above 65 in group 1 (2 weeks), 68 in group 2 (4 weeks) and 75 in group 3 (8 weeks). These values increased until euthanasia, being above 77 in group 1, 72 in group 2 and 73 in group 3, all of which were considered to be very high. All of the means, as well as the standard deviations and statistical significance, both for the initial ISQ (ISQ SURG) and for the day of euthanasia (ISQ EUT) in the different study groups are shown in Table 1, Table 2 and Table 3. Figure 3 compares the ISQ SURG and EUT values according to the time of sacrifice and for each surface type.

There was a statistically significant difference between the three surfaces in the ISQ EUT value in groups 2 and 3, with the highest values for the SLA^®^ surface. For group 1, euthanised at 2 weeks, the mean TSI TSQ for the 3 surfaces did not differ significantly, although it did for groups 2 and 3. At 4 weeks, the ContacTi^®^ surface presented mean values significantly lower than those for SLA^®^, but not significantly different compared to those for SLActive^®^ (Tukey, *p* = 0.112). At 8 weeks, ContacTi^®^ presented significantly lower mean values than those for SLA^®^ and SLActive^®^, but the latter do not show significant differences between them (Tukey, *p* = 0.874). (Table 2)

### 2.3. Bone-Implant Contact (BIC)

Statistical methods were used to detect the presence of outliers. As a result, it was found that one of the samples from the ContacTi^®^ surface showed a very different value for the BIC variable relative to the rest of the samples. The statistical tests revealed this outlier; yet, given the impossibility of detecting the likely cause that could justify this value, it was decided to carry out the data analysis both with the outlier and without it. Due to the resulting data being significantly different, it was concluded that the outlier should be removed. Table 4 summarises the mean values of the % Bone-Implant Contact (BIC) once the outlier had been removed for all three study groups, according to the time of sacrifice. The three surfaces showed their lowest values when analysed at 2 weeks (group 1). The %BIC values then increased significantly over time to reach very high values, i.e., greater than 81%, at 4 weeks (no significant differences were found between the values reached for each surface). At 8 weeks, the SLActive^®^ and ContacTi^®^ surfaces achieved maximum values of −87.1 and 90.02, respectively, with no significant differences between both values (*p* = 0.066), although there were significant differences with respect to the value of the SLA^®^ surface of −77.9. The ContacTi^®^ surface obtained the highest values of all three measurements at 2, 4, and 8 weeks with respect to the other two. Figure 4 shows the mean %BIC values for the different surfaces at the different time points of the study.

Comparing the different time points of the study for each surface, significant differences were detected in all cases (*p* = 0.003 for SLA^®^ and *p* < 0.001 for SLAactive^®^ and ContactTi^®^), with a %BIC significantly higher at 4 and 8 weeks compared to 2 weeks on the three surfaces.

### 2.4. Correlation

The correlation between RFA and %BIC was studied for the three groups at the different time points in the study. No significant correlation was found between ISQ and BIC (*p* > 0.05 for all associations) for each implant surface and time (Table 5).

## 3. Discussion

The aim of this study was to compare the behaviour of three different implant surfaces in an animal model by utilising histomorphometric (BIC) and implant stability (ISQ) data, as well as to establish the possible statistical correlation between both variables.

### 3.1. ISQ and BIC Values for the Three Types of Surfaces

For successful implants, it has been reported that the ISQ values vary in a range between 57 and 82, with a mean of 69 after one year [21]. In the present study, the ISQ SURG means for the day of surgical insertion ranged between 65 and 81 ISQ, with means greater than 71 for the three surfaces, and increased until the time of sacrifice. These values are considered high and are related to the stiffness of the bone-implant interface, determining the primary stability of the implant, which depends on factors such as the surgical technique, the macroscopic design of the implant and the thickness and density of the peri-implant bone [10,13,22].

When assessing the ISQ-EUT value, which is collected at the time of sacrifice, the concept of secondary stability is already a factor, beginning with the process of bone remodelling, when osteoclasts begin the process of bone resorption that ensures primary stability, with the result being the formation of new bone around the implant surface [23]. In this study, the ISQ-EUT value of the three surfaces at different times is considered very high, i.e., above 72. These values varied according to the time of sacrifice, so that at two weeks, no significant differences were observed between values for each surface, whereas at 4 and 8 weeks, the value of the ContacTi^®^ surface was significantly lower than those of the other two, which may be due to different factors that influence this value, such as the macroscopic design of the implant or the milling protocol used. The ISQ-EUT values for the different surfaces ranged between 72.5 and 81.6 ISQ, which are high values, and were also quite stable at the different time points of the study, which suggest the optimal behaviour of the three surfaces in terms of primary and secondary stability. The absolute value of the different measurements obtained is less important due to the difficulty involved in knowing what percentages of primary and secondary stability are involved in the ISQ value obtained at a given time. In addition, as time passes, it is important to understand that ISQ values tend to equalise independently of the initial values obtained [14,24].

BIC is a histomorphometric value widely used in the literature to measure osseointegration [5,16]. The results of this study showed very high values of %BIC in the groups at 4 and 8 weeks, compared with the values obtained in the 2 weeks group, at a significant difference. The results were especially high for the bioactive surfaces (SLActive^®^ and ContacTi^®^) in the group at 8 weeks, and the difference was significant with respect to the SLA^®^ surface, but not between the surfaces.

The main factor that influences the apposition of bone on implants is considered today to be related to the characteristics of its surface [25,26]. It has been clearly demonstrated that rough surfaces tend to yield greater bone formation when compared with smooth surfaces, with a positive correlation between the BIC value and surface roughness [27,28]. However, surface characteristics such as energy and wettability also influence BIC values. In this sense, Buser et al. [29] compared the SLA^®^ and SLActive^®^ surfaces, finding that the BIC was higher with the SLActive^®^ surface until 4 weeks, while it equalised for both surfaces towards 8 weeks. Our study obtained similar results in that the %BIC value increased throughout the different time points in the study, such that the values obtained at 4 and 8 weeks were significantly higher than those obtained in the 2-week group. However, our study differs in that in the group sacrificed at 8 weeks, both SLActive^®^ and ContacTi^®^ showed values that were significantly higher than those for SLA^®^. This can be justified by the fact that SLActive^®^ and ContacTi^®^ are bioactive surfaces that would present a faster rate of bone tissue growth around the implants, as has been demonstrated for both surfaces [5,30].

### 3.2. Correlation between ISQ and %BIC Values

The hypothesis that a greater amount of bone tissue in contact with the implant surface (elevated BIC) should be related to an increase in the stiffness of the bone-implant interface, and therefore, of its stability measured with RFA, seems reasonable. In an attempt to validate this hypothesis, numerous articles have been published in recent years trying to assess this correlation. Thus, Meredith et al. [31] in 1997 in rabbits, Nkenke et al. [20] in 2003 and Gedrange et al. [19] in 2005 in cadavers, Schliephake [6] in 2006 in dogs, Scarano et al. [32] in 2006 and Huwiler et al. [33] in 2007 in humans, Ito et al. [17] in 2008 in pigs, Strnad et al. [24] in 2008 in dogs, Abrahamsson et al. [34] in 2009 in dogs, Degidi et al. [35] in 2009 in humans, Jun et al. [36] in 2010 in cadavers, Stadlinger et al. [37] in 2009 in pigs, Blanco et al. [38] in 2011 in rabbits, Abdel-Haq et al. [39] in 2011 in sheep, Park et al. [40] in 2012 in rabbits, Manresa et al. [41] in 2014 in dogs, Dagher et al. [42] in 2014 and Soares et al. [43] in 2015 in rabbits, Acil et al. [44] in 2016 in pigs, Gehrke et al. [45] in 2016 in rabbits and Chang et al. [46] in 2016 in dogs, have analysed whether the RFA is valid for assessing the stability of the bone-implant complex by correlating the stability values (ISQ) with the histomorphometric variables, especially the BIC variable, in a range of study models. However, the results have been contradictory: while some studies find a clear correlation [19,20,24,32,38,44,46], many others find the opposite [16,17,31,34,35,36,37,40,41,42,43]. The cause of the difference in the results is due, first, to the significant heterogeneity among the studies conducted, which makes them difficult to compare. Specifically, clinical studies have been conducted on humans, cadavers and different types of animal models (dogs, pigs, sheep, rabbits, etc.). In addition, the anatomical locations studied vary (mandible, maxillary, tibia, femur, etc.). However, the analysed histomorphometric variables vary according to the parameters (bone-implant contact, BIC; bone volume density, BVD; bone area density, BAD; etc.), as well as the device used to measure the RFA, when different versions of Ostell^®^ (Ostell, Ostell Mentor, Ostell ISQ) are used. All of these factors create difficulty in drawing clear and reliable conclusions from the set of published articles referenced above.

In the present study, we did not obtain a statistical correlation between ISQ values and %BIC values. The three surfaces studied achieved very high results for both variables but without a correlation between the two factors. Different explanations can be suggested for this result, such as the histomorphometric variable undergoing important changes in the first weeks as a consequence of the process of bone remodelling, which induces the formation of immature bone that decreases the bone-implant union achieved during the phase of primary stability.

Some studies have pointed to the fact that all implants, regardless of their initial stability, show a tendency to achieve similar levels of stability over time, independent of the BIC presented [14,31]. Based on this fact, perhaps an explanation for the lack of an RFA and BIC correlation found in our study would be that the degree of bone-implant union may not reflect the degree of stiffness of the bone surrounding the implant, which is what RFA actually measures. This stiffness depends on factors such as the percentage of bone-implant contact and the thickness of the bone layer or bone density around the implant, among other factors. In support of this, the literature has described the fact that implants with different BIC values can have similar stability values or that implants with similar BIC values can have different ISQ values [47,48].

Three published studies investigated the same animal model as ours, minipigs. Ito et al. [17] did not find a correlation between the two variables in a study on tibias using implants covered with hydroxyapatite, nor did Stadlinger et al. [37] on jaw implants coated with collagen/chondroitin sulphate, both coinciding with our results. However, Acil et al. [44] found a moderate correlation in their study on the frontal bone of domestic pigs, although measurements were performed immediately after insertion of the implants.

All three surfaces studied demonstrated very high RFA and BIC values in the different phases of the study, with the highest values exhibited by the bioactive surfaces, SLActive^®^ and the new surface ContacTi^®^. The latter shows excellent behaviour in animal models, favouring faster and better bone apposition on the implant surface. However, it was not possible to find a correlation between the two variables studied, which reflects that the high stability achieved in this study, as measured by RFA, and therefore the increase in bone anchoring cannot be adequately explained by the high values of bone-implant anchorage obtained. Perhaps the fact that the histomorphometric variables used are collected from two-dimensional histological sections led to this lack of correlation, thus it is possible that the results obtained here could be better explained by new studies using an artificial model and 3-dimensional microcomputed tomography. Such an approach may shed more light on the correlation between the implant-bone union, by recording values based on the three true dimensions in the system, known as 3D BIC, and the RFA, which is a little explored field that requires future research [49].

This study didn’t use a human model for ethical reasons, which can be a limitation, however, the animal model in minipigs is sufficiently validated in the literature for both histomorphometric and implant stability studies. The significant differences in the methodologies used by the different authors makes it difficult to normalise and compare the data obtained in our study with the data reported in the others, thus we believe it is necessary to standardise methodological criteria before conducting new studies so that the variables involved may be properly assessed.

## 4. Materials and Methods

Three different surface treatments were compared from the perspective of implant histomorphometric and stability characters. SLA^®^ (Institut Straumann AG, Basel, Switzerland): A rough titanium surface was obtained by sandblasting with corundum particles and subsequent etching with a mixture of HCl/H_2_SO_4_ at an elevated temperature for several minutes. SLActive^®^ (Institut Straumann AG, Basel, Switzerland): Another surface treatment (SLActive^®^ Institut Straumann AG, Basel, Switzerland), similar to the previous one was employed, followed by submerging the implant in a nitrogen protective atmosphere and continuously stored in an isotonic NaCl solution until the implant enters the bone bed, to prevent the passive elements from coming into contact with the surface of the implant, maintaining the active and hydrophilic nature of the surface. The performance of both surfaces was evaluated, with comparisons made between these surfaces and the performance of a third, which is a new bioactive surface obtained by a two-step technique. This new surface is fabricated by initially blasting the surface with particles of alumina on the surface of the titanium and then by applying a subsequent thermochemical treatment, in which the metal was immersed in a solution of 10 mL of 5 M NaOH at 60 °C for 24 h and subsequent washing with distilled water and drying at 40 °C for 24 h. Finally, the surfaces were subjected to a heat treatment in a tubular oven at 600 °C for one hour, yielding the surface type known as ContacTi^®^ (Klockner S.A. Via Augusta, 158-9ª planta-08006 Barcelona SPAIN & SOADCO S.L. Avgda. Fiter i Rossell 4 Bis local 2 AD700 Escaldes-Engordany ANDORRA) [30]. The previously stated data shows that the new ContacTi surface presents an improved performance in the osseointegration process when compared to the classical surface, accelerating and achieving better results in the histomorphometric parameters. These results could evidence that the apatite layer formed over the implant surface when in contact with the surrounding bone has an adequate surface roughness achieved by the formation of bone like material on its surface, promoting osteoblastic migration, bonding, proliferation, and differentiation [7,30].

An animal model was used in this study. Twelve female 6-year-old minipigs were selected at the Centralized Animal Experimentation Service of the University of Cordoba (Registry number 20-08-15-293) according to policy 202/707UE, which regulates the use of animals (date: 24-Aug-2015) This study was previously approved by the Ethics Committee of Experimentation of the University of Seville (date: 13-Nov-2010) and was in compliance with all of the requirements and regulations for animal experimentation, in accordance with current regulations in Spain and the European Union.

The implants used with the SLA^®^ and SLActive^®^ surface treatments were 4.1 mm in diameter and 8 mm in length with a polished neck measuring 1.8 mm in height. The surface treatment ContacTi^®^ was used in implants of 4 mm in diameter and 8 mm in length with a polished neck measuring 1.5 mm in length. All implants were placed by a surgeon who is highly experienced in the management of the implants used. This animal model and procedure was validated in previous studies [50,51].

The study consisted of two surgeries: in the first intervention, the posterior teeth, premolars and molars, were extracted from the jaws of all of the pigs by raising a full-thickness flap, followed by an osteotomy to leave a suitable area for the placement of implants in the second phase. An antibiotic and analgesic treatment were used to prevent postoperative pain and inflammation. After a 6-month healing period, three implants were placed in the maxilla of each pig, one for each surface studied, using a semi-submerged technique and in accordance with the manufacturer’s instructions. The polished collar was placed supracrestally, with a distance between the implants of at least 4 mm. Healing abutments of 2-mm height were placed, and interrupted suturing of the incision was performed using 4/0 synthetic polyamide (Supramid^®^), with a not submerged technique.

After placement of the implants and before suturing, the initial stability of each implant was recorded using the resonant frequency analysis (RFA) system. For this, the Osstell ISQ^®^ device (Ostell AB, Göteborg, Sweden) was employed. This device is a wireless system in which a metal pin with a magnetised tip, the Smartpeg, is screwed into the implant at a force of 4–6 N/cm ^2^. The probe of the transducer is directed towards the pin without touching it and is excited by magnetic pulses so that the resonance frequency is recorded by the device and expressed as an implant stability quotient (ISQ value), with units ranging from 1 to 100. Two records were taken for each implant, recording the mean of the two, such that the recording probe was always placed for each implant in the same way.

The animals were divided into three groups according to the time at which they were euthanised: Group 1, at 2 weeks of implant placement; Group 2, at 4 weeks; and Group 3, at 8 weeks. The minipigs were euthanized by an overdose of sodium pentothal via carotid perfusion, complemented by a mixture of 5% glutaraldehyde and 4% formaldehyde, at pH 7.2. Immediately after sacrifice, the implants were exposed to record the stability value (ISQ) by RFA (Osstell ISQ device). The same procedure noted above was followed for the day of placement.

For the histomorphometric study, sections, including the implant, alveolar bone and mucosa, were obtained from each animal. The samples were processed using the technique described by Donath and Breuner [52]: the EXAKT system (Exact Vertriebs, Norderstedt, Germany) was used to process the sections with methyl methacrylate (MMA) as the inclusion medium. An Exakt 310^®^ oscillating saw was used to delimit the implant and surrounding bone with a maximum thickness of 4 mm to ensure proper fixation. The samples were submerged for fixation in a 10% buffered formalin solution for a minimum of two weeks until subsequently processed. The samples were finally dehydrated by increasing concentrations of ethanol (70, 80, 96, 100, and 100%) in one-day steps and were subsequently infiltrated and embedded in the methyl methacrylate resin (MMA) (Technovit 7200 VLC, Kulzer-Heraus, Germany).

Using a diamond band saw, each polymerized block was cut by following the axial direction of the implant. The saw was irrigated to avoid overheating and deterioration of the tissues surrounding the implant, operated at a maximum rotational speed of the band saw and minimum forward advancement. All sections were polished to an optical finish with an automatic polisher (Exakt 400 CS, Exakt, Germany).

The polished sections were examined by scanning electron microscopy (SEM) (JSM-6400, JEOL, Japan) using retro-dispersed electrons to differentiate the surface bone in detail. The resolution used in the study was 8 nm.

The histomorphometric variable studied was the BIC (bone-implant contact), or the amount of mineralized bone that is in intimate contact with the surface of the implant. Being a parameter of length, the BIC is therefore a bidimensional static variable.

Statistical analysis was carried out using the IBM SPSS 23.0 statistical package for Windows. The means and standard deviations were analyzed by groups, performing multiple comparison tests to determine the statistical significance (established at *p* < 0.05). The means and standard deviations were determined overall and by groups. For a comparison of the numerical variables between the groups/time, two-way ANOVA was applied after verifying the normality of the data using the Shapiro-Wilk test and the homogeneity of the variance using Levene’s test. When these tests were significant, multiple comparisons, DMS (groups of the same dimension), and Tukey HSD (groups of slightly different dimension) were used.

The correlation between the quantitative variables (ISQEut and BIC) was performed with a Spearman’s correlation coefficient because the trends observed between the variables were rarely linear. Scatter plot representations were used, as well as the Dixon and Grubb’s Tests, for the detection of outliers.

## 5. Conclusions

In this study, which was performed on an animal model (minipig jaws), the novel surface ContacTi^®^ showed remarkable results in terms of the osseointegration process, achieving excellent histomorphometric and implant stability values in the different times of the study. No statistical correlation was found between the RFA values (ISQ values) and %BIC. These data coincide with those found in other previous studies, both in humans and in animal models, and there is significant methodological variability that creates difficulty in comparing the results and obtaining reliable conclusions.

## Figures and Tables

**Figure 1 ijms-20-02307-f001:**
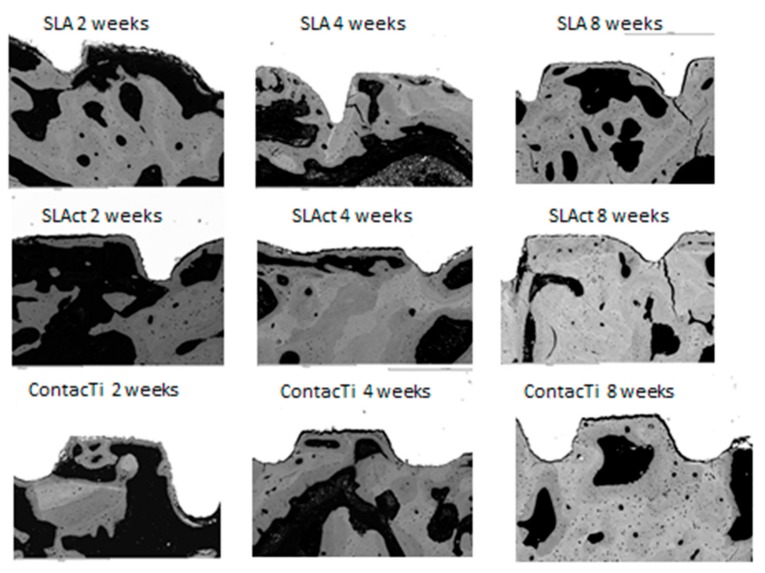
Representative histological images of the three surfaces (ContacTi^®^, SLA = SLA^®^, SLAct = SLActive^®^) in the jaw at the different time points of the study (2, 4, and 8 weeks). Magnification 75×.

**Figure 2 ijms-20-02307-f002:**
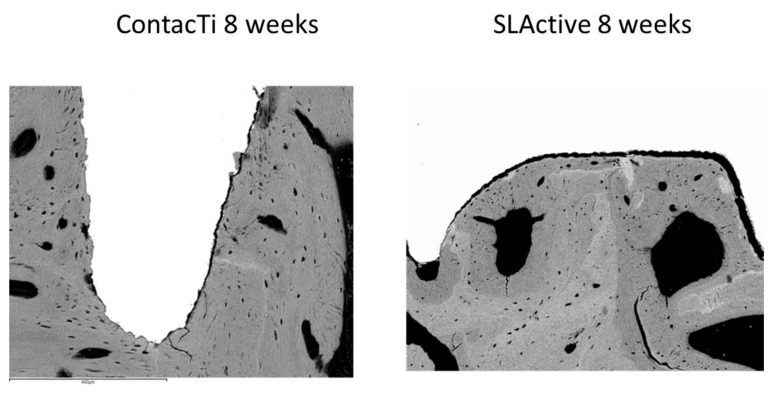
Histological images of the ContacTi^®^ and SLActive^®^ surfaces obtained by SEM in a maxillary section at 8 weeks, at 150× magnification.

**Figure 3 ijms-20-02307-f003:**
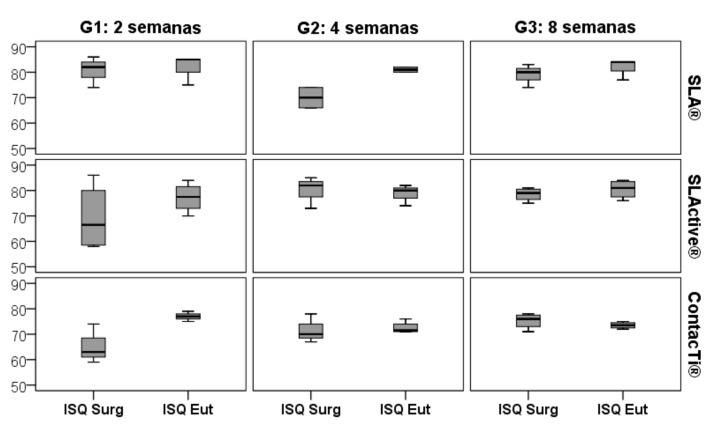
Boxplot for comparison of the ISQ SURG and ISQ EUT values by surface studied throughout the different time points of the study.

**Figure 4 ijms-20-02307-f004:**
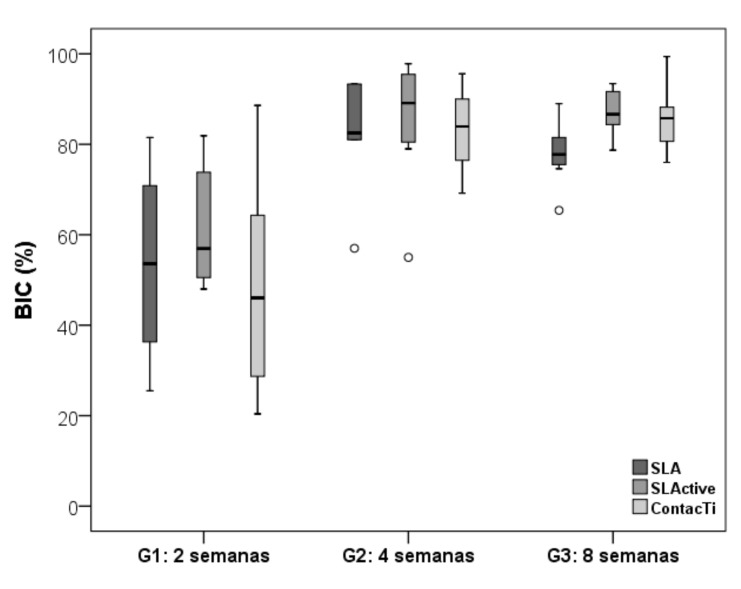
Mean values of the bone-implant contact, BIC (%), for the three surfaces at the different time points in the study. With the box diagram are determined the outliers (circle) that go out of the standard (extreme values).

**Table 1 ijms-20-02307-t001:** Relative means of the ISQ value in the measurements performed in group 1 (euthanasia at 2 weeks), both on the day of surgery (ISQ SURG) and at the time of euthanasia (ISQ EUT), for each of the three groups.

GROUP 1	ISQ SURG		ISQ EUT (2 Weeks)	
	*N*	Mean ± SD	*N*	Mean ± SD
SLA^®^	4	81.25 ± 5.12	3	81.67 ± 5.77
SLActive^®^	4	69.25 ± 13.35	4	77.25 ± 5.85
ContacTi^®^	3	65.33 ± 7.77	3	77.00 ± 2.00
*p*		0.124		0.467

surg = surgery; SD = standard deviation; eut = euthanasia.

**Table 2 ijms-20-02307-t002:** Relative means of the ISQ value in the measurements performed in group 2 (euthanasia at 4 weeks), both on the day of surgery (ISQ SURG) and at the time of euthanasia (ISQ EUT), for each of the three groups. The lowercase letters show in standard form (columns) the differences in pairs between the means by the Tukey method after detecting statistical significance in the ANOVA.

GROUP 2	ISQ SURG		ISQ EUT (4 Weeks)	
	*N*	Mean ± SD	*N*	Mean ± SD
SLA^®^	4	68.00^B^ ± 4.69	2	81.00^a^ ± 1.41
SLActive^®^	4	76.00 ± 9.49	3	78.67^ab^ ± 4.16
ContacTi^®^	4	71.25 ± 4.72	4	72.50^bB^ ± 2.38
*p*		0.285		0.030

surg = surgery; SD = standard deviation; eut = euthanasia.

**Table 3 ijms-20-02307-t003:** Relative means of the ISQ value in the measurements performed in group 2 (euthanasia at 8 weeks), both on the day of surgery (ISQ SURG) and at the time of euthanasia (ISQ-TSU), for each of the three groups. The lowercase letters show in standard form (columns) the differences in pairs between the means by the Tukey method after detecting statistical significance in the ANOVA.

GROUP 3	ISQ SURG		ISQ EUT (8 Weeks)	
	*N*	Mean ± SD	*N*	Mean ± SD
SLA^®^	4	79.25^A^ ± 3.77	3	81.67^a^ ± 4.04
SLActive^®^	4	78.50 ± 2.65	4	80.50^a^ ± 3.70
ContacTi^®^	4	75.25 ± 3.10	4	73.50^bAB^ ± 1.29
*p*		0.226		0.015

surg = surgery; SD = standard deviation; eut = euthanasia.

**Table 4 ijms-20-02307-t004:** Mean %BIC values at different points in the study and the absolute means.

BIC (%)	Group 1: EUT at 2 Weeks		Group 2: EUT at 4 Weeks		Group 3: EUT at 8 Weeks		*p*
	*N*	Mean ± SD	*N*	Mean ± SD	*N*	Mean ± SD	
SLA^®^	8	53.56B ± 19.96	6	81.60A ± 13.31	8	77.99bA ± 6.89	0.003
SLActive^®^	8	61.56B ± 13.39	8	85.36A ± 13.98	8	87.18aA ± 4.93	<0.001
ContacTi^®^	6	62.35B ± 20.01	8	88.25A ± 17.67	8	90.0258abA ± 9.90	<0.001
*p*		0.493		0.847		0.020	

EUT = euthanasia; SD = Standard deviation; ^a, b^—different letters indicate significantly different means between different surface treatments according to the DMS test; ^A, B^—different letters indicate significantly different means between groups according to the Tukey HSD test.).

**Table 5 ijms-20-02307-t005:** Correlation between ISQEut and BIC, and between ISQEut and BAT for each implant system (surface) and time (group).

		G1: 2 Weeks	G2: 4 Weeks	G3: 8 Weeks	G1: 2 Weeks	G2: 4 Weeks	G3: 8 Weeks
Surface		ISQ Eut vs. BIC (%)			ISQ Eut vs. BAT (%)		
SLA^®^	Spearman’s rho	0.828	0.943	0.000	0.828	0.894	−0.621
	*p*	0.052	0.057	1.000	0.052	0.106	0.188
	*N*	6	4	6	6	4	6
SLActive^®^	Spearman’s rho	0.683	0.956	−0.146	0.634	−0.598	−0.537
	*p*	0.062	0.053	0.729	0.091	0.210	0.170
	*N*	8	6	8	8	6	8
ContacTi^®^	Spearman’s rho	−0.717	−0.206	0.244	−0.478	−0.466	−0.098
	*p*	0.109	0.625	0.560	0.338	0.245	0.818
	*N*	6	8	8	6	8	8

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
