# Peer review of "Influence of a Novel Surface of Bioactive Implants on Osseointegration: A Comparative and Histomorfometric Correlation and Implant Stability Study in Minipigs"

_ijms, 2019, doi:10.3390/ijms20092307_

Round 1
Reviewer 1 Report
The reviewer found it confusing for the materials and methods to be listed at the end of the manuscript. Unless it is a requirement by the journal I suggest moving the methodology after the introduction.
More details on how the new implant surface (ContacTi) are necessary. For example, how is it produced? What is the merit of new surface in comparison to well established surfaces?
More details are needed regarding the surgical procedure for placing the implant, were the implants submerged or not? I am assuming the implants were not submerged, but this details has to be added to the text.
The limitations of the study have to be well-outlined within the discussion.
Author Response
Letter of response to the comments of the editors - Referee 1
First of all, we would like to thanks you for your comments about our job. We appreciate them and we have tried to make all the changes you proposed. In the following writing, we answered one by one to your observations and at the end we attach the article with all the modifications requested by the respective reviewers.
Referee 1:
English language and style
( ) Extensive editing of English language and style required
( ) Moderate English changes required
(x) English language and style are fine/minor spell check required
( ) I don't feel qualified to judge about the English language and style
Yes | Can be improved | Must be improved | Not applicable | |
Does the introduction provide sufficient background and include all relevant references? | (x) | ( ) | ( ) | ( ) |
Is the research design appropriate? | ( ) | (x) | ( ) | ( ) |
Are the methods adequately described? | ( ) | (x) | ( ) | ( ) |
Are the results clearly presented? | ( ) | (x) | ( ) | ( ) |
Are the conclusions supported by the results? | ( ) | (x) | ( ) | ( ) |
Comments and Suggestions for Authors
The reviewer found it confusing for the materials and methods to be listed at the end of the manuscript. Unless it is a requirement by the journal I suggest moving the methodology after the introduction.
I agree with you; materials and methods should be after introduction, but it is a requirement by the journal and all authors respect it.
More details on how the new implant surface (ContacTi) are necessary. For example, how is it produced? What is the merit of new surface in comparison to well established surfaces?
I followed your instructions
More details are needed regarding the surgical procedure for placing the implant, were the implants submerged or not? I am assuming the implants were not submerged, but this details has to be added to the text.
I followed your instructions
The limitations of the study have to be well-outlined within the discussion.
I followed your instructions.
Reviewer 2 Report
The study evaluated osseointegration of a novel surface treatment of titanium implant and compared it to established surface treatments. Bone implant contact (BIC) analysis and implant stability, measured with resonant frequency analysis (RFA, with ISQ units) were compared after implantation in minipigs.
My comments and suggestions for improvement of the manuscript are as follows:
Abstract:
Line 15: The sentence is too long. Please break it up to make it reader friendly.
Purpose: The objective of this study was to assess the influence of a novel surface of dental
implants (ContacTi®) on the osseointegration process in a minipig model, comparing this surface
with other existing surfaces on the market (SLA® and SLActive®) by employing bone implant
contact (BIC) analysis and implant stability, measured with resonant frequency analysis (RFA, with
ISQ units)
Several other long sentences permeate the manuscript and make it painful to read. Check Line 199-203
Introduction:
Line 65: I believe the sentence should read as follows…..(my addition is in bold font, underlined)
RFA has proven to be a useful tool for assessing the evolution of implant osseointegration because it allows for a clinical measurement of the quality of the bone-implant interface immediately after insertion (primary stability), as well as biological anchorage or secondary stability that occurs in the following weeks [15].
Results
Figure titles: Please be consistent with the language of communication. Change Semanas to weeks
In Table 3 the values for sample size (n) and mean +SD have been flipped. Please correct the error.
Line 155: Is the maximum %BIC value for SLActive negative 87.1? If not please remove the minus sign in front of -87.1. Same issue applies at line 157 for SLA surface of -77.9.
4. Table 4: A two way Anova with independent variables of treatment and time is appropriate for
this data but the authors report a one-way Anova test. Please correct.
5. Column 4 of the table provides an absolute mean value for bone contact over the three time periods. I
don’t believe that is a meaningful result and I will recommend that it should be removed.
Discussion
Line 207 Please re-write these sentences for readability ” The results of this study showed very high values of %BIC in the groups at 4 and 8 weeks compared with the values obtained in the 2 weeks group, at a significant difference. The results were especially high for the bioactive surfaces (SLActive® and ContacTi®) in the group at 8 weeks, and the difference was significant with respect to the SLA® surface but not between the surfaces. (?what surfaces)
Materials and Methods
Line 291: Please provide information on how long etching was performed.
Line 294: The explanation of the technique for obtaining a hydrophilic surface is inadequate. Please explain what passive elements are prevented from contacting the implant surface after etching. How is the implant transferred from the nitrogen environment to the implant bed to prevent contamination?
Line 311: Implant length is 8 mm
Line 322: Are the healing abutments really 0 mm?
Conclusion
The objectives of the study were;
to assess the influence of a novel surface of dental implants (ContacTi®) on the osseointegration process in a minipig model, comparing this surface with other existing surfaces on the market (SLA® and SLActive®) by employing bone implant contact (BIC) analysis and implant stability, measured with resonant frequency analysis (RFA, with ISQ units)
and also evaluate whether a correlation exists between BIC and RFA values.
The conclusion only addresses the second objective but not the first. Please expand the conclusions to address your first objective.
Thank you.
Author Response
Letter of response to the comments of the editors - Referee 2
First of all, we would like to thanks you for your comments about our job. We appreciate them and we have tried to make all the changes you proposed. In the following writing, we answered one by one to your observations and at the end we attach the article with all the modifications requested by the respective reviewers.
Revisor 2:
English language and style
( ) Extensive editing of English language and style required
( ) Moderate English changes required
(x) English language and style are fine/minor spell check required
( ) I don't feel qualified to judge about the English language and style
Yes | Can be improved | Must be improved | Not applicable | |
Does the introduction provide sufficient background and include all relevant references? | (x) | ( ) | ( ) | ( ) |
Is the research design appropriate? | (x) | ( ) | ( ) | ( ) |
Are the methods adequately described? | ( ) | (x) | ( ) | ( ) |
Are the results clearly presented? | ( ) | (x) | ( ) | ( ) |
Are the conclusions supported by the results? | ( ) | ( ) | (x) | ( ) |
Comments and Suggestions for Authors
The study evaluated osseointegration of a novel surface treatment of titanium implant and compared it to established surface treatments. Bone implant contact (BIC) analysis and implant stability, measured with resonant frequency analysis (RFA, with ISQ units) were compared after implantation in minipigs.
My comments and suggestions for improvement of the manuscript are as follows:
Abstract: Line 15: The sentence is too long. Please break it up to make it reader friendly. Purpose: The objective of this study was to assess the influence of a novel surface of dental implants (ContacTi®) on the osseointegration process in a minipig model, comparing this surface with other existing surfaces on the market (SLA® and SLActive®) by employing bone implant contact (BIC) analysis and implant stability, measured with resonant frequency analysis (RFA, with ISQ units)
I followed your instructions
Several other long sentences permeate the manuscript and make it painful to read. Check Line 199-203 I followed your instructions
Introduction:
Line 65: I believe the sentence should read as follows…..(my addition is in bold font, underlined) RFA has proven to be a useful tool for assessing the evolution of implant osseointegration because it allows for a clinical measurement of the quality of the bone-implant interface immediately after insertion (primary stability), as well as biological anchorage or secondary stability that occurs in the following weeks [15].
I followed your instructions
Results
Figure titles: Please be consistent with the language of communication. Change Semanas to weeks I followed your instructions
In Table 3 the values for sample size (n) and mean +SD have been flipped. Please correct the error. Line 155: Is the maximum %BIC value for SLActive negative 87.1? If not please remove the minus sign in front of -87.1. Same issue applies at line 157 for SLA surface of -77.9. I followed your instructions
4. Table 4: A two way Anova with independent variables of treatment and time is appropriate for this data but the authors report a one-way Anova test. Please correct.
I followed your instructions
5. Column 4 of the table provides an absolute mean value for bone contact over the three time periods. I don’t believe that is a meaningful result and I will recommend that it should be removed. I followed your instructions
Discussion
Line 207 Please re-write these sentences for readability ” The results of this study showed very high values of %BIC in the groups at 4 and 8 weeks compared with the values obtained in the 2 weeks group, at a significant difference. The results were especially high for the bioactive surfaces (SLActive® and ContacTi®) in the group at 8 weeks, and the difference was significant with respect to the SLA® surface but not between the surfaces. (?what surfaces) I followed your instructions
Materials and Methods
Line 291: Please provide information on how long etching was performed.
I followed your instructions
Line 294: The explanation of the technique for obtaining a hydrophilic surface is inadequate. Please explain what passive elements are prevented from contacting the implant surface after etching. How is the implant transferred from the nitrogen environment to the implant bed to prevent contamination? I followed your instructions
Line 311: Implant length is 8 mm I followed your instructions
Line 322: Are the healing abutments really 0 mm? I followed your instructions
Conclusion
The objectives of the study were; to assess the influence of a novel surface of dental implants (ContacTi®) on the osseointegration process in a minipig model, comparing this surface with other existing surfaces on the market (SLA® and SLActive®) by employing bone implant contact (BIC) analysis and implant stability, measured with resonant frequency analysis (RFA, with ISQ units) and also evaluate whether a correlation exists between BIC and RFA values.
The conclusion only addresses the second objective but not the first. Please expand the conclusions to address your first objective. I followed your instructions
Thank you.